# A visual circuit uses complementary mechanisms to support transient and sustained pupil constriction

William Thomas Keenan[1†], Alan C Rupp[1†], Rachel A Ross[2,3], Preethi Somasundaram[4], Suja Hiriyanna[5], Zhijian Wu[5], Tudor C Badea[5], Phyllis R Robinson[4], Bradford B Lowell[6,7,8], Samer S Hattar[1,9*]

[1]Department of Biology, Johns Hopkins University, Baltimore, United States; [2]Department of Psychiatry, Beth Israel Deaconess Medical Center, Harvard Medical School, Boston, United States; [3]Department of Psychiatry, Massachusetts General Hospital, Boston, United States; [4]Department of Biological Sciences, University of Marlyand, Baltimore, United States; [5]National Eye Institute, National Institutes of Health, Bethesda, United States; [6]Division of Endocrinology, Diabetes, and Metabolism, Beth Israel Deaconess Medical Center, Harvard Medical School, Boston, United States; [7]Department of Medicine, Beth Israel Deaconess Medical Center, Harvard Medical School, Boston, United States; [8]Program in Neuroscience, Harvard Medical School, Boston, United States; [9]Department of Neuroscience, Johns Hopkins University, Baltimore, United States

*For correspondence: shattar@ jhu.edu

†These authors contributed equally to this work

Competing interests: The authors declare that no competing interests exist.

**Abstract** Rapid and stable control of pupil size in response to light is critical for vision, but the neural coding mechanisms remain unclear. Here, we investigated the neural basis of pupil control by monitoring pupil size across time while manipulating each photoreceptor input or neurotransmitter output of intrinsically photosensitive retinal ganglion cells (ipRGCs), a critical relay in the control of pupil size. We show that transient and sustained pupil responses are mediated by distinct photoreceptors and neurotransmitters. Transient responses utilize input from rod photoreceptors and output by the classical neurotransmitter glutamate, but adapt within minutes. In contrast, sustained responses are dominated by non-conventional signaling mechanisms: melanopsin phototransduction in ipRGCs and output by the neuropeptide PACAP, which provide stable pupil maintenance across the day. These results highlight a temporal switch in the coding mechanisms of a neural circuit to support proper behavioral dynamics.

## Introduction

Environmental light influences a variety of subconscious physiological functions, including circadian photoentrainment, light modulation of sleep/mood, and the pupillary light response (PLR). These diverse effects of light are all mediated by a small subpopulation of retinal output neurons called intrinsically photosensitive retinal ganglion cells (ipRGCs) (*Altimus et al., 2008*; *Göz et al., 2008*; *Güler et al., 2008*; *Hatori et al., 2008*; *LeGates et al., 2012*; *Lupi et al., 2008*; *Tsai et al., 2009*). Even in the vast array of environmental light conditions, subconscious visual behaviors are remarkable for their rapid induction and stable maintenance throughout the day. However, how the ipRGC circuit achieves rapid and stable control of visual behaviors remains uncertain.

Multiple photoreceptive systems participate in the ipRGC circuit, including their endogenous melanopsin-based phototransduction and indirect synaptic input from the classical rod and cone

**eLife digest** The retina is the part of our eye that detects light and sends visual information to the brain. There are several different types of light-sensitive cell in the retina that perform different roles. For example, retinal cells called intrinsically photosensitive retinal ganglion cells (or ipRGCs for short) rapidly respond to the intensity of background light and regulate the size of the pupils to control how much light enters the eyes. These cells receive information from other light-sensitive cells in the retina called rods and cones. There are at least two mechanisms that ipRGCs may use to relay information to the brain: one uses a protein called PACAP, while the other involves a molecule called glutamate. However, it is still not clear which mechanisms are actually used by ipRGCs, or when they might use them.

Like other mammals, mice can rapidly reduce the size of their pupils when they are suddenly exposed to a bright light. Keenan, Rupp et al. investigated how ipRGCs control the size of the pupils in mice that had been genetically engineered to lack different components of the visual system. Mutant mice that lacked rod cells or were unable to produce glutamate in their ipRGCs failed to reduce the size of their pupils when a bright light was switched on. In contrast, other mutant mice that were unable to produce a light-sensitive pigment in their ipRGCs showed a normal response initially, but had trouble keeping their pupils small if the light stayed on for a longer period of time. The same was true for mice that were missing the PACAP protein in their ipRGCs.

These findings show that ipRGCs use different systems to quickly alter the size of the pupil in response to sudden changes in light level and then to maintain the size of the pupil over a longer period of time. Further work is needed to find out if ipRGCs use the same mechanisms to control the other behaviors they influence, such as mood and sleep patterns.

photoreceptors (*Hattar et al., 2003*; *Panda et al., 2003*). Each photoreceptive system presumably encodes a unique aspect of the light environment, but to date no consensus exists on the photoreceptive mechanisms supporting ipRGC-dependent behaviors. Several studies using a variety of methods have proposed competing models arguing for the predominance of cone-based (*Allen et al., 2011*; *Butler and Silver, 2011*; *Dkhissi-Benyahya et al., 2007*; *Lall et al., 2010*) or rod-based (*Altimus et al., 2010*; *McDougal and Gamlin, 2010*) synaptic input to ipRGCs and their behavioral responses. Additionally, it has been suggested that melanopsin mediates persistent light detection in ipRGCs (*Altimus et al., 2008*; *Gooley et al., 2012*; *Lupi et al., 2008*; *Mrosovsky and Hattar, 2003*; *Zhu et al., 2007*) because melanopsin phototransduction is relatively slow to initiate but stable for minutes to hours (*Berson et al., 2002*; *Gooley et al., 2012*; *Wong, 2012*). However, animals lacking melanopsin still retain sustained light responses in ipRGCs and their central targets (*Schmidt et al., 2014*; *van Diepen et al., 2013*; *Wong, 2012*) and relatively normal circadian photoentrainment (*Panda et al., 2002*; *Ruby et al., 2002*) and PLR (*Lucas et al., 2003*; *Xue et al., 2011*). In total, it remains unclear how ipRGCs utilize each distinct photoreceptive input, especially across the environmental range of light intensities and durations.

ipRGCs must faithfully relay information about the light environment to the brain. Many neurons, including ipRGCs, release multiple neurotransmitters, a classical neurotransmitter and one or more neuropeptides (*Vaaga et al., 2014*). However, methods to evaluate mammalian cotransmitter systems in vivo in real time are lacking. ipRGCs contain the principal excitatory neurotransmitter glutamate and the neuropeptide PACAP (pituitary adenylyl cyclase-activating polypeptide) (*Engelund et al., 2010*; *Hannibal et al., 2002*). Recent studies have suggested that glutamate is the predominant regulator of ipRGC-dependent behaviors, including circadian photoentrainment and the PLR (*Delwig et al., 2013*; *Gompf et al., 2015*; *Purrier et al., 2014*). By comparison, animals lacking PACAP or its receptors show at best minor deficits in circadian photoentrainment and the PLR (*Beaulé et al., 2009*; *Colwell et al., 2004*; *Engelund et al., 2012*; *Kawaguchi et al., 2010*, *2003*). This difference in outcomes between glutamate and PACAP has led to the conclusion that PACAP is dispensable and serves primarily as a modulator of glutamatergic signaling (*Chen et al., 1999*). It remains puzzling why ipRGCs, like many other neuronal cell types, would possess two distinct neurotransmitters.

To date, the precise behavioral contributions of rod, cone, and melanopsin input or their output neurotransmitters glutamate and PACAP to visual behaviors across time are essentially unknown. Here, we have systematically addressed the behavioral contributions of all three photoreceptive inputs and both neurotransmitter outputs of ipRGCs, and how these change with time. To do so, we have silenced each individual photoreceptor or neurotransmitter component of ipRGCs, and in multiple combinations, while measuring pupil size across environmental light intensities and time domains. We have taken advantage of the fact that the PLR provides the unique opportunity to dissect the precise temporal dynamics of inputs and outputs of the ipRGC circuit in a behaving animal. This study reveals how ipRGC circuit dynamics in vivo support pupil regulation across time and provides insights into ipRGC regulation of other subconscious visual behaviors.

## Results

### ipRGC behavioral responses are composed of both transient and sustained phases

To measure ipRGC responses in real time, we measured the pupillary light response (PLR). Importantly, we used a novel experimental setup that mimics environmental light using overhead light with spectral composition similar to daylight in an unanesthetized mouse (*Figure 1A* and *Figure 1—figure supplement 1*), unlike previous studies that used monochromatic light delivered to a single eye (*Delwig et al., 2013*; *Gooley et al., 2012*; *Güler et al., 2008*; *Kawaguchi et al., 2010*; *Lall et al., 2010*; *Lucas et al., 2003*).

Following light onset, we observed rapid pupil constriction that is maintained for the duration of the 30-s recording (*Figure 1B*), with greater constriction under higher light intensities (*Figure 1D*). Previous studies have noted a PLR decay during a sustained light stimulus lasting minutes (*Gooley et al., 2012*; *Loewenfeld, 1993*; *McDougal and Gamlin, 2010*), prompting us to systematically monitor the pupil across a range of times and light intensities. We observed a decay in pupil constriction over time that reached a new steady state (*Figure 1C*), resulting in two phases in the PLR: transient and sustained (mean intensity to reach 50% constriction ($EC_{50}$) for transient PLR = 0.53 lux, sustained PLR = 7.9 lux)(*Figure 1D,E*). Because pupil constriction itself lowers the amount of light reaching the retina and therefore limits the drive to continued pupil constriction, the PLR is a form of negative feedback. To test if PLR decay is a consequence of negative feedback, we measured the effect of negative feedback both computationally and experimentally, and found that it has little role in PLR decay (*Figure 1—figure supplement 2*). Furthermore, we observed full PLR decay at dim light intensities ($\leq$1 lux) within the first 5 min of light stimulation (*Figure 1C,F*), but full maintenance of pupil constriction at high light intensities ($\geq$1000 lux), with apparently slower decay rates at higher light intensities (half-life: ~2–5 min, *Figure 1F*). These results suggest that ipRGCs possess temporally distinct inputs and/or outputs for transient and sustained signaling.

### Transient input to ipRGCs is mediated by rods

To identify the photoreceptor(s) inputs that contribute to transient ipRGC responses (*Figure 2A*), we tested the PLR in mutant mouse lines that lack the function of a single photoreceptor type, leaving the function of the other photoreceptors intact (*Table 1*, for references on production and initial characterization of each line); we refer to these lines as cone knockout, rod knockout, and melanopsin knockout mice. To corroborate our findings, we tested a variety of mutant mouse lines that silence each photoreceptor type in unique ways (*Table 1*).

Importantly, these mutant mouse lines have been extensively tested for visual function (*Alam et al., 2015*; *Altimus et al., 2010*; *Biel et al., 1999*; *Cahill and Nathans, 2008*; *Calvert et al., 2000*; *Naarendorp et al., 2010*; *Nathan et al., 2006*; *Zhao et al., 2014*). Rod sensitivity and function is unchanged in cone mutant animals and cone sensitivity and function is unchanged in rod mutant animals (*Alam et al., 2015*; *Altimus et al., 2010*; *Biel et al., 1999*; *Cahill and Nathans, 2008*; *Calvert et al., 2000*; *Naarendorp et al., 2010*; *Nathan et al., 2006*). Electrophysiological recordings of ipRGCs show functional rod input in cone mutants and functional cone input in rod mutants (*Zhao et al., 2014*). Additionally, all of the photoreceptor mutant lines we used have similar pupil sizes in darkness (*Figure 2—figure supplement 1*). Therefore, these mouse lines allow precise

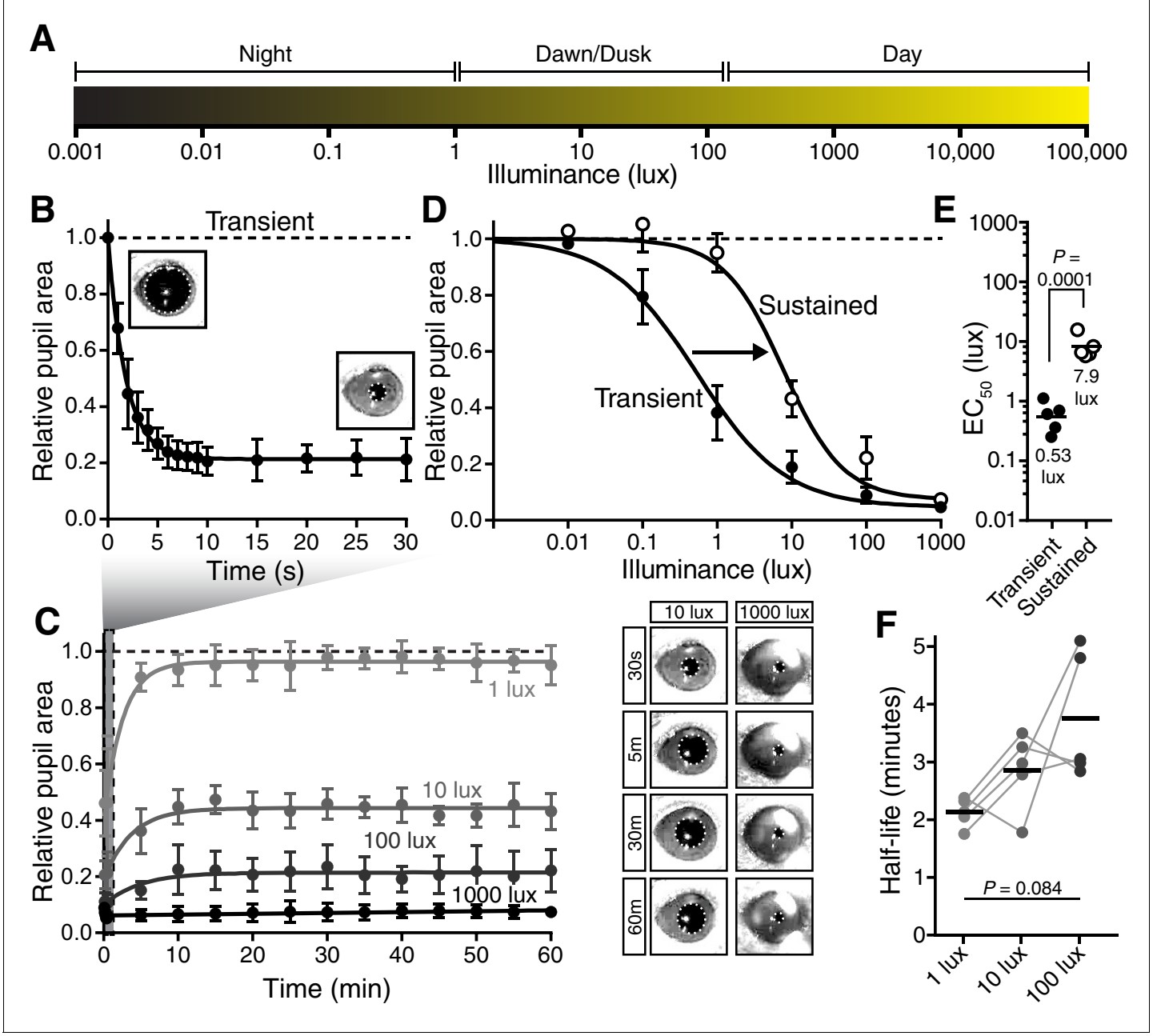

**Figure 1.** The pupillary light response contains two phases: transient and sustained. (A) Approximate light intensity ranges (lux) at different times of day. (B) Transient constriction in response to a 10 lux overhead stimulus (mean ± SD). Boxes contain representative pupil images at time 0 and 30 s. (C) Continued monitoring of pupil constriction from b for 60 min of continuous light at 5 min intervals with representative images. (D) Intensity-response curve for transient and sustained constriction (30 s and 60 min, respectively). Data fit with a sigmoidal curve ($n$ = 5, mean ± SD). (E) Light intensity required for half-maximal constriction ($EC_{50}$) determined for both transient and sustained phases of the PLR. $EC_{50}$ extracted from the sigmoidal curve fits for each mouse (points are individual mice, line is mean). Statistical significance determined with a student's $t$ test. (F) Half-life of PLR decay at 1, 10, and 100 lux. Statistical significance determined by main effect of light intensity from one-way ANOVA. See also *Figure 1—figure supplement 1*, *Figure 1—figure supplement 2*.

The following figure supplements are available for figure 1:

**Figure supplement 1.** Experimental setup and light stimulus details.

**Figure supplement 2.** Negative-feedback model of PLR decay.

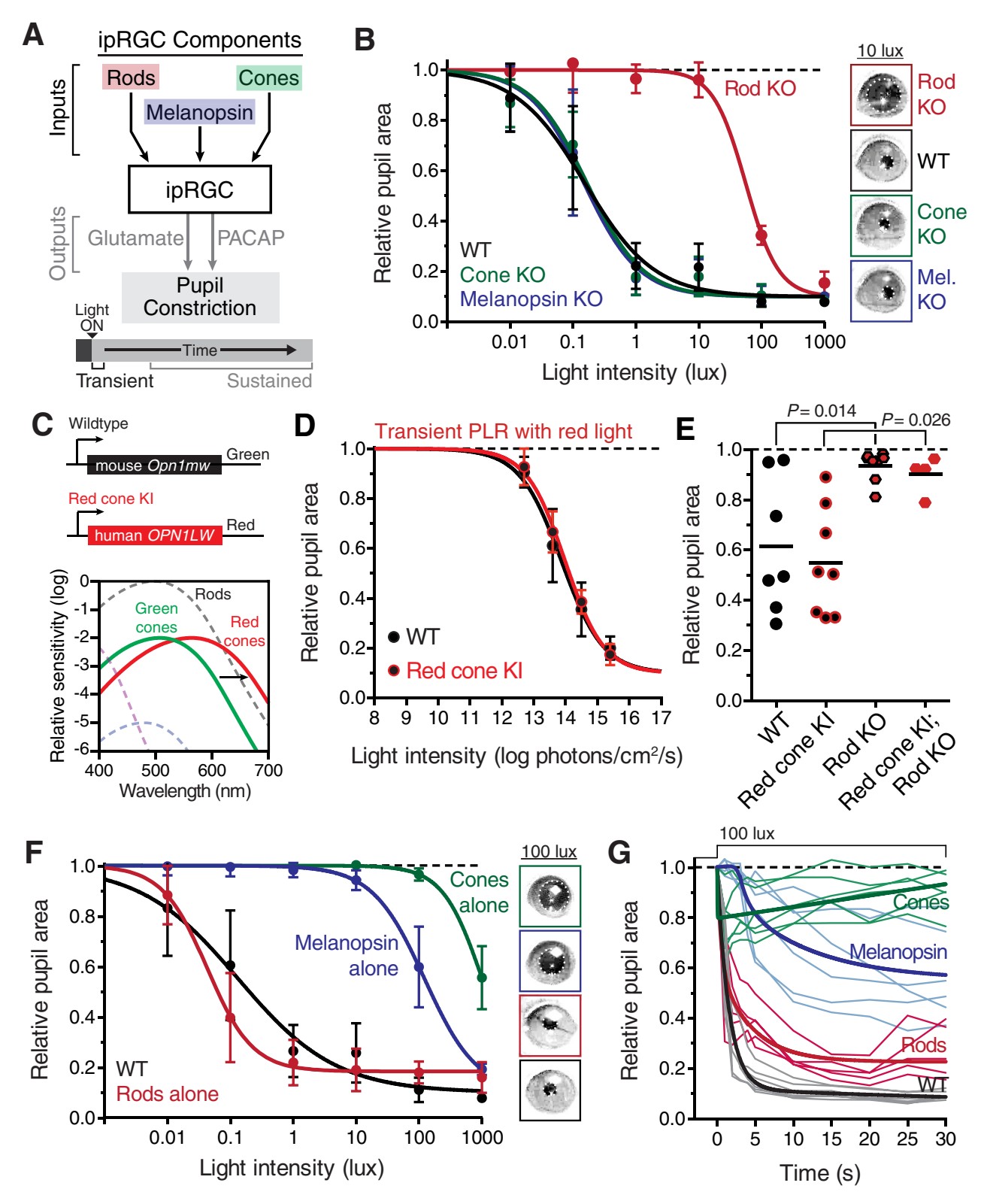

**Figure 2.** Transient input to ipRGCs is mediated by rods. (**A**) Diagram of ipRGC behavioral circuit. (**B**) Intensity-response curves of the PLR in each of the photoreceptor mutant mouse lines (mean ± SD): wildtype (*n* = 6), Rod KO (*Gnat1⁻/⁻ n* = 6), Melanopsin KO (*Opn4⁻/⁻ n* = 8), and Cone KO (*Gnat2⁻/⁻ n* = 7). Representative pupil images for each mouse line at 10 lux. (**C**) Gene schematic comparison of endogenous mouse M-cone allele and human red cone knock-in allele as well as the spectral sensitivity shift observed. Notice that cones are more sensitive to red light in Red cone KI line. (**D**) The PLR to

*Figure 2 continued*

red light (626-nm LED) is identical in mice with cones that are more sensitive to red light (Red cone KI, *n* = 6) compared to littermate WT (*n* = 5), mean ± SD. (E) Removing rod function abolishes the PLR in response to red light (626-nm LED), even in mice with cones with enhanced sensitivity to red light. WT *n* = 7, Red cone KI (*Opn1mw^{red}*) *n* = 8, Rod KO (*Gnat1^{-/-}*)⁻ *n* = 8, Red cone KI; Rod KO (*Gnat1^{-/-}; Opn1mw^{red}*) *n* = 4. Light intensity is 14.3 log photons/cm²/s. (F) Intensity-response curves in mutant mice with each photoreceptor isolated (Rod-only: *Cnga3^{-/-}; Opn4^{-/-}* *n* = 6)(Cone-only: (*Gnat1^{-/-}; Opn4^{-/-}* *n* = 6)(Mel.-only: *Gnat1^{-/-}; Gnat2^{-/-}* *n* = 7) Data is mean ± SD, statistical significance determined using a one-way ANOVA with Sidak's post-test. (right) Representative pupil images at 100 lux. (G) Kinetics of transient pupil constriction (100 lux) in mice with only rod, cone, or melanopsin function, same genotypes and number of animals as in F. Traces of individual mice are shown behind curve-fits. One-phase decays were fit to all except cone-only which was fit with a two-phase decay due to its rapid pupil decay within 30 s. Melanopsin-only kinetic fit was offset from 0 by 3 s to account for delay in constriction. See also *Figure 2—figure supplements 1–5*.

The following figure supplements are available for figure 2:

**Figure supplement 1.** Dark-adapted pupil sizes of photoreceptor mutant mouse lines used.

**Figure supplement 2.** Rods are required for the transient phase of the PLR.

**Figure supplement 3.** Melanopsin is not required for transient PLR in response to environmentally relevant overhead light.

**Figure supplement 4.** Rod input to the transient PLR is influenced by cones.

**Figure supplement 5.** Melanopsin can drive rapid constriction at high light intensities.

separation of rod, cone, and melanopsin activation while leaving the function of the other photoreceptors intact.

When we tested the transient PLR of rod, cone, and melanopsin mutant mice, we found that both cone and melanopsin knockout mice were identical to wildtype in both sensitivity and kinetics (*Figure 2B* and *Figure 2—figure supplement 2B*). Despite previous reports of melanopsin requirement for the transient PLR (*Lucas et al., 2003*), we find that melanopsin is dispensable for the PLR when using more environmentally relevant stimuli (*Figure 2—figure supplement 3*). In contrast, rod knockout mice displayed no pupil constriction until the light intensity becomes relatively bright (i.e. >10 lux, *Figure 2B*), despite the normal spatial vision in rod knockout mice at these moderate light

**Table 1.** Description of photoreceptor mutant mouse lines used.

| Mouse line | Genotype | Effect on retinal function | Citations |
|---|---|---|---|
| Rod KO | *Gnat1^{-/-}* | No rod phototransduction | (*Calvert et al., 2000*) |
| Rod-DTA | rdta | No rod cell bodies; cones present early in life | |
| Cone KO1 | *Cnga3^{-/-}* | No cone phototransduction | (*Biel et al., 1999*) |
| Cone KO2 | *Gnat2^{cpfl3/cpfl3}* | No cone phototransduction | (*Chang et al., 2006*) |
| Cone-DTA | h.red DT-A | Ablation of all M cones; >95% loss of S cones | (*Soucy et al., 1998*) |
| Melanopsin KO | *Opn4^{-/-}* | No melanopsin phototransduction | (*Lucas et al., 2003*) |
| Cone-only | *Gnat1^{-/-}; Opn4^{-/-}* | No rod/melanopsin phototransduction | |
| Rod-only 1 | *Cnga3^{-/-}; Opn4^{-/-}* | No cone/melanopsin phototransduction | |
| Rod-only 2 | *Gnat2^{-/-}; Opn4^{-/-}* | No cone/melanopsin phototransduction | |
| Rod-only 3 | h.red DT-A; *Opn4^{-/-}* | No cone cells nor melanopsin phototransduction | |
| Melanopsin-only 1 | *Gnat1^{-/-}; Cnga3^{-/-}* | No rod/cone phototransduction | |
| Melanopsin-only 2 | *Gnat1^{-/-}; Gnat2^{-/-}* | No rod/cone phototransduction | |
| Melanopsin-only 3 | rdta; h.red DT-A | No rod or cone cell bodies | |
| Red cone KI | *Opn1mw^{red}* | Cones have shifted sensitivity to red | (*Smallwood et al., 2003*) |
| Red cone KI; Rod KO | *Opn1mw^{red};Gnat1^{-/-}* | Cones have shifted sensitivity to red, no rod phototransduction | |

intensities (*Alam et al., 2015*). To corroborate these results, we tested three different cone mutant lines and two different rod mutant lines with distinct mutations and observed virtually identical results: cone mutants are similar to wildtype and rod mutants have severe transient sensitivity deficits (*Figure 2—figure supplement 2C,D*).

These results are surprising given previous proposals that cones are important for transient ipRGC responses, including acute changes in pupil size (*Allen et al., 2011*; *Dkhissi-Benyahya et al., 2007*; *Gooley et al., 2012, 2010*; *Ho Mien et al., 2014*; *Kimura and Young, 2010, 1999*; *Lall et al., 2010*; *Spitschan et al., 2014*; *van Oosterhout et al., 2012*). Therefore, we sought to acutely modulate cone activity using a previously characterized mouse line that expresses the human 'red' opsin (*OPN1LW*) in place of the mouse 'green' opsin (*Opn1mw*) (Red cone KI), making cones the only photoreceptors with enhanced sensitivity to red light (*Lall et al., 2010*) (*Figure 2C*). We found that these mice have identical transient PLR in response to red light as wildtype (*Figure 2D*), indicating that acute cone modulation does not affect the overall magnitude of the PLR. Furthermore, crossing this line to a rod knockout line abolishes the PLR in response to red light (*Figure 2E*). These results show that rods are the predominant photoreceptor inputs for transient PLR at low to moderate light intensities, even in a mouse line with sensitized cones.

To evaluate the inputs contributed by each photoreceptor in isolation to the PLR, we generated double mutants lacking the function of two photoreceptor types, resulting in mice with only rods (Rods alone), only cones (Cones alone) or only melanopsin (Melanopsin alone) (*Table 1*). We found that the only photoreceptors capable of recapitulating the wildtype PLR are rods. Mice with only rod function had identical light sensitivity as wildtype and a similar rapid induction of pupil constriction (*Figure 2F,G*), though their ability to maintain stable pupil sizes in bright light was slightly diminished (*Figure 2G*). We corroborated the sufficiency of rods using three different mouse lines (*Figure 2—figure supplement 4*). Interestingly, while two of the lines were nearly identical to wildtype, one line had similar sensitivity, but altered kinetics, suggesting that cones might regulate rod signaling dynamics.

In marked contrast to rod input, cone and melanopsin inputs were severely deficient in mediating the transient PLR (*Figure 2F,G*). Animals with melanopsin alone retained a normal PLR at bright light intensities (*Figure 2F*), as seen previously (*Gooley et al., 2012*; *Lucas et al., 2001*; *Xue et al., 2011*), with sensitivity that is indistinguishable from rod knockouts (*Figure 2—figure supplement 5*), though they had relatively sluggish kinetics (*Figure 2G*). In contrast, cone-only animals had minimal PLR (*Figure 2F*), resulting in a further sensitivity deficit compared to rod knockout and melanopsin-only animals (*Figure 2—figure supplement 5*). Additionally, cone input decayed rapidly (*Figure 2G*), presumably due their robust light adaptation properties.

Collectively, these results show that rods serve as the primary input to ipRGCs for transient PLR responses, especially at low to moderate light intensities. At bright light intensities, additional input originates predominantly from melanopsin phototransduction.

## Glutamaterigic output provides precise and rapid transient signaling

To investigate how ipRGCs relay transient light detection to the brain, we tested the transient PLR in mice lacking glutamatergic neurotransmission in ipRGCs (*Opn4$^{Cre/+}$* ; *Slc17a6$^{fl/fl}$*, also known as *Vglut2$^{fl/fl}$*) or mice lacking PACAP in ipRGCs (*Opn4$^{Cre/+}$* ; *Adcyap1$^{fl/-}$*) (*Figure 3A* and *Table 2*). See *Figure 3—figure supplement 2* for details on design of the conditional PACAP allele (*Adcyap1$^{fl}$*).

Though ipRGC glutamate knockout mice (*Opn4$^{Cre/+}$* ; *Slc17a6$^{fl/fl}$*) exhibited a small decrease in resting pupil size (*Figure 3—figure supplement 1*) (*Delwig et al., 2013*), we observed that they had minimal transient PLR at all light intensities (*Figure 3B–E*), with more robust PLR at very bright light intensities (*Figure 3—figure supplement 3*), in agreement with previous studies (*Delwig et al., 2013*; *Purrier et al., 2014*). This indicates that ipRGC glutamatergic neurotransmission is a critical transient signal for the PLR. Presumably, the residual transient response is PACAPergic.

In contrast to ipRGC glutamate knockout mice, ipRGC PACAP knockout mice had no deficits in transient PLR sensitivity or kinetics (*Figure 3B–E*), as observed previously (*Kawaguchi et al., 2010*), suggesting that glutamate is sufficient for the entirety of the transient PLR. Additionally, these results show that any potential modulation of glutamatergic signaling by PACAP (*Chen et al., 1999*; *Toda and Huganir, 2015*) is dispensable for the transient PLR. Together, these data derived from retinal mutants for photoreceptors and neurotransmitters identify rods as the principal input and glutamate as the principal output of ipRGC-mediated transient PLR signaling.

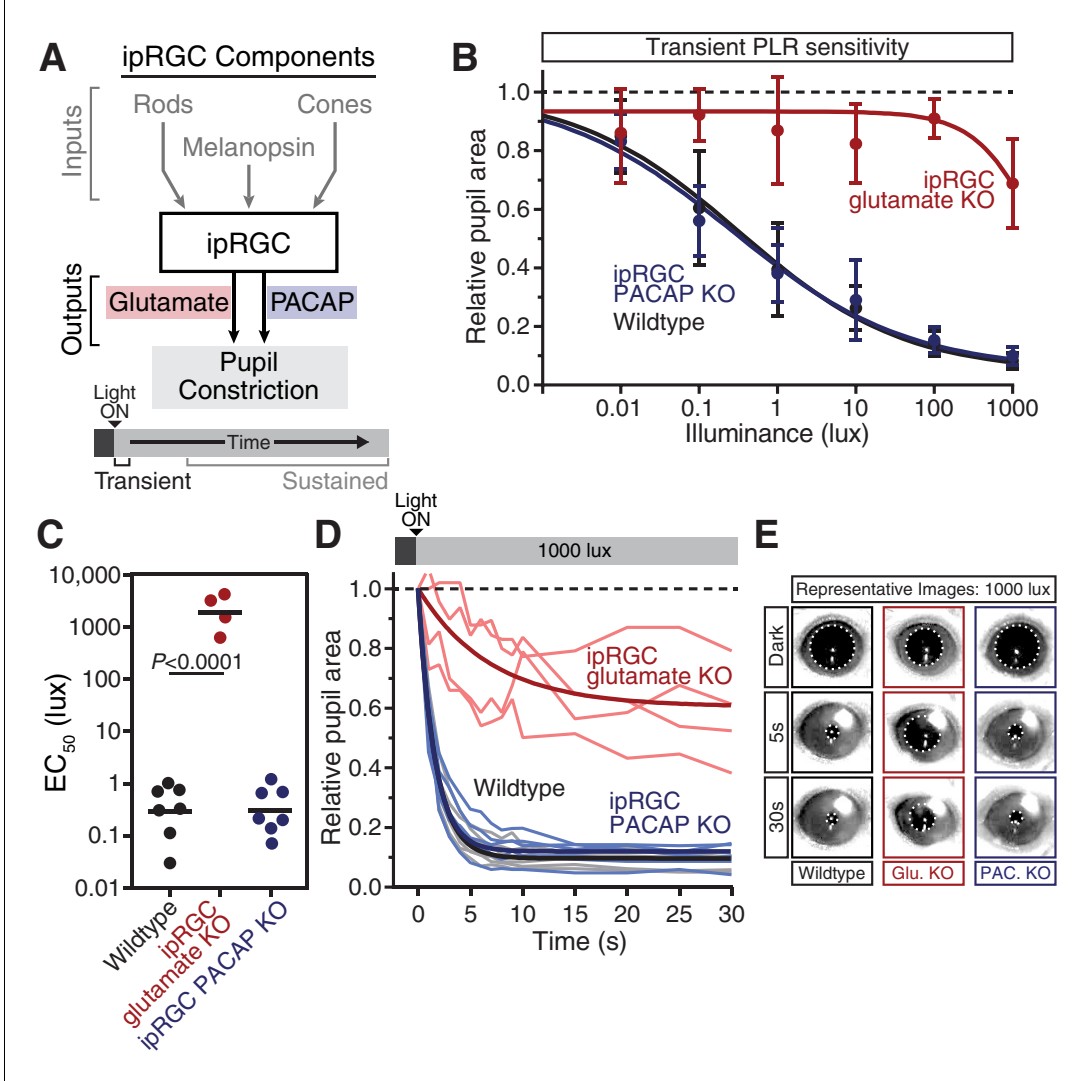

**Figure 3.** Glutamaterigic output provides precise and rapid transient signaling. (**A**) Diagram of ipRGC behavioral circuit. (**B**) Intensity-response curves of the PLR in each of the neurotransmitter mutant mouse lines (Wildtype $n = 6$) (ipRGC glu. KO: $Opn4^{Cre/+}$ ; $Slc17a6^{fl/fl}$ $n = 4$) (ipRGC PACAP KO: $Opn4^{Cre/+}$ ;$Adcyap1^{fl/-}$ $n = 6$)(mean ± SD). (**C**) Sensitivity (EC$_{50}$) in each of the mutant lines. Statistical significance determined by one-way ANOVA with Sidak's post-test. (**D**) Kinetics of transient pupil constriction (1000 lux) in mice lacking glutamatergic or PACAPergic neurotransmission. Traces of individual mice are shown behind one-phase decay curve-fits. Half-lives: Wildtype (1.1 s), ipRGC glu. KO (4.8 s), ipRGC PACAP KO (1.1 s). (**E**) Representative pupil images at 5 s and 30 s post-illumination (1000 lux). *Figure 3—figure supplements 1–3*.

The following figure supplements are available for figure 3:

**Figure supplement 1.** Dark-adapted pupil sizes of neurotransmitter mutant lines used.

**Figure supplement 2.** Description of conditional PACAP allele.

**Figure supplement 3.** PACAP can drive significant constriction within 30s of high light onset.

## Melanopsin/rod synergy supports PLR under sustained conditions

Since wildtype responses decay over time (***Figure 1***), we next asked how ipRGC inputs and outputs drive the PLR across longer times (***Figure 4A***). Strikingly, when we measured the sustained PLR in melanopsin knockout mice, which have a normal transient PLR (***Figure 2B***), there was virtually no pupil constriction (***Figure 4B***), even at bright light intensities (up to 10,000 lux, ***Figure 4—figure***

**Table 2.** Description of neurotransmitter mutant mouse lines used.

| Mouse line | Genotype | Effect on retinal function | Citations |
|---|---|---|---|
| Melanopsin-Cre | $Opn4^{Cre/+}$ | Cre expression in ipRGCs | (*Ecker et al., 2010*) |
| *Slc17a6*-flox | $Slc17a6^{fl/fl}$ | Exon 2 flanked by loxP sites | (*Hnasko et al., 2010*) |
| ipRGC glutamate KO | $Opn4^{Cre/+}$; $Slc17a6^{fl/fl}$ | Silences ipRGC glutamatergic release | |
| PACAP KO | $Adcyap1^{-/-}$ | Whole animal PACAP removal | (*Hamelink et al., 2002*) |
| PACAP-flox | $Adcyap1^{fl/fl}$ | Exon 2 flanked by loxP sites | See *Figure 3—figure supplement 2* |
| ipRGC PACAP KO | $Opn4^{Cre/+}$ ; $Adcyap1^{fl/-}$ | Silences ipRGC PACAP release | |

*supplement 1A*). We observed that melanopsin knockout mice lose pupil constriction in minutes (half-life: ~4 min, *Figure 4C*), similar to the wildtype PLR decay rate at lower light intensities (WT half-life range: ~2–4 min at 1–100 lux, *Figure 1F*). This suggests that melanopsin phototransduction maintains robust light input in ipRGCs during the day (*Figure 4—figure supplement 1B*), after rods adapt to background light.

The severe deficits we observed in the sustained PLR in melanopsin knockout mice raised the possibility that these animals may have developmental deficits that affect their signaling (*Rao et al., 2013*; *Renna et al., 2011*). To directly address this issue, we rescued ipRGC function in adult melanopsin knockout mice using either chemogenetics or restoration of melanopsin expression. Using our mouse line with Cre introduced into the melanopsin locus ($Opn4^{Cre/Cre}$) and a Cre-dependent chemogenetic DREADD virus (AAV2-hSyn-DIO-hM3D(G$_q$)-mCherry) (*Figure 4—figure supplement 2*), we administered the selective DREADD agonist CNO (*Armbruster et al., 2007*) and observed robust and sustained pupil constriction for at least one hour (*Figure 4D*). This result demonstrates that ipRGCs and their downstream circuits remain competent for sustained signaling in melanopsin knockout mice. Furthermore, we acutely restored melanopsin in the majority of ipRGCs of melanopsin-Cre knockout mice ($Opn4^{Cre/Cre}$) using a virus that expresses melanopsin in a Cre-dependent manner (*Figure 4E* and *Figure 4—figure supplement 2C–E*, AAV2-CMV-DIO-mRuby-P2A-Melanopsin-FLAG). Following melanopsin restoration, we observed a rescue of the sustained PLR (*Figure 4F*). These results demonstrate for the first time that the effect of melanopsin loss can be rescued in adulthood, indicating that melanopsin-based light detection is directly required for ipRGCs to signal sustained PLR.

Surprisingly, although melanopsin is required for sustained signaling, we found that melanopsin signaling could not fully recapitulate the sustained PLR. Despite the observation that the sustained PLR is normal at bright light intensities in melanopsin-only mice, these mice had a sensitivity deficit compared to wildtype (*Figure 4G*). Notably, we observed that rod knockout mice display an identical sensitivity deficit as melanopsin-only (*Figure 4G* and *Figure 4—figure supplement 3*), indicating that rods contribute to sustained ipRGC signaling. This indicates that at intermediate intensities, both rod and melanopsin signaling cooperate to sustain the PLR.

As with the transient PLR, we found that cone knockout mice had no deficit in sustained PLR (*Figure 4G*). Again, multiple independent mouse lines corroborate these conclusions (*Figure 4—figure supplement 3*). Furthermore, we found that rods alone could drive the remainder of the sustained PLR in melanopsin knockout mice (*Figure 4—figure supplement 4A*), whereas cone-only mice had no sustained PLR (*Figure 4—figure supplement 4B*).

These results show that melanopsin signaling dominates sustained light input to ipRGCs, but rods, which are thought to be nonfunctional under continuous bright light, are intimately involved in supporting the sustained PLR. Notably, rod contributions to the sustained PLR occur predominantly at light intensities above their presumed saturation (~40 lux), showing that rods are indeed capable of contributing to visual function above previously defined limits (*Alam et al., 2015*; *Altimus et al., 2010*; *Naarendorp et al., 2010*). Therefore, sustained ipRGC responses are not a simple consequence of a single photoreceptive system, but instead require rod/melanopsin synergy for highest sensitivity.

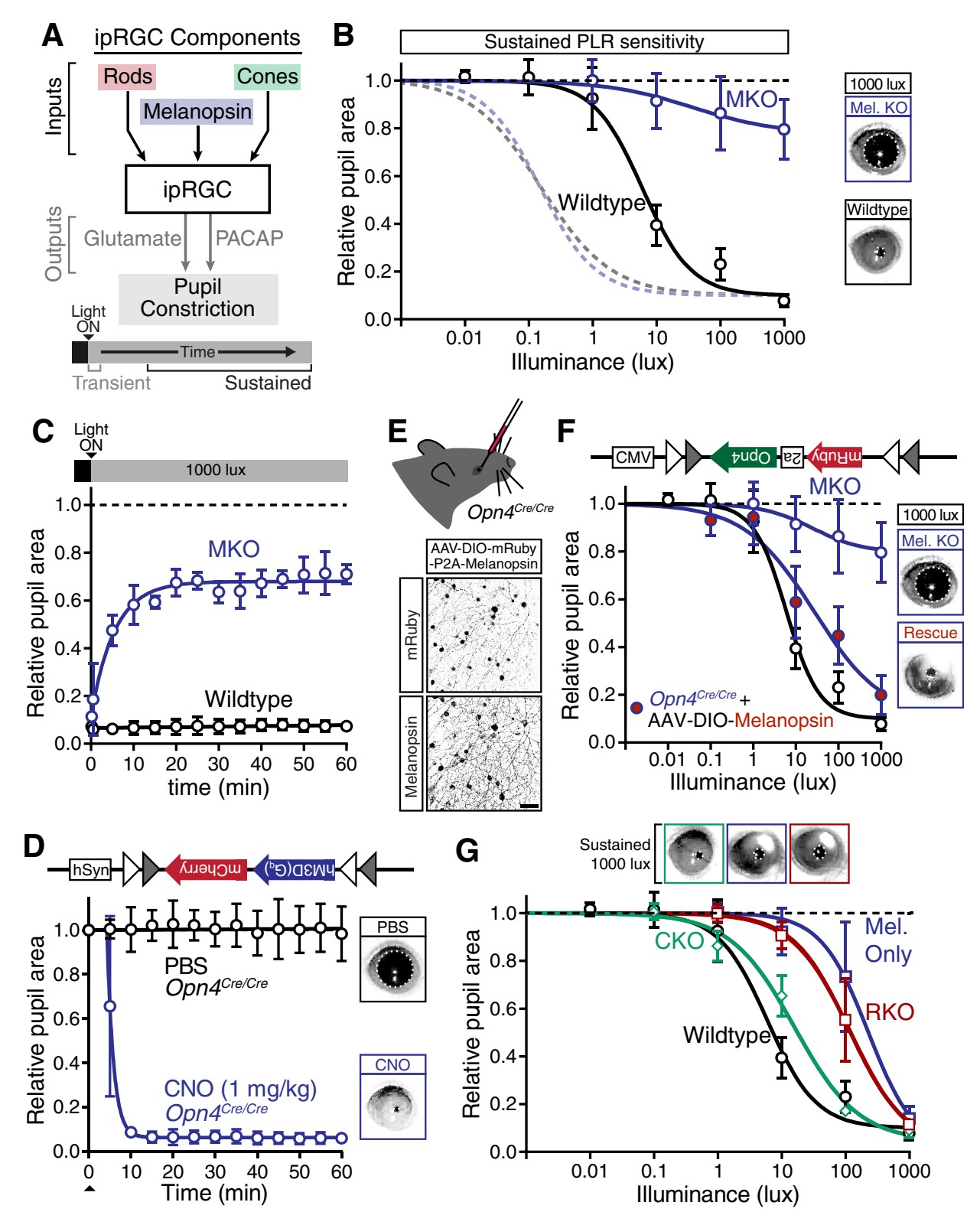

**Figure 4.** Melanopsin/rod synergy supports PLR under persistent conditions. (**A**) Diagram of ipRGC behavioral circuit. (**B**) Intensity-response curves for wildtype and melanopsin knockout mice (*Opn4$^{-/-}$*): transient (dotted lines for reference) and sustained (60 min: solid lines) (WT *n* = 6, *Opn4$^{-/-}$ n* = 12). (right) Representative pupil images under 1000 lux persistent light. (**C**) 60-min time course of pupil constriction under constant light (1000 lux). Data fit with a one-phase association curve (WT *n* = 5, *Opn4$^{-/-}$ n* = 7). (**D**) Sustained pupil constriction monitored every 5 min for 1 hr in melanopsin knockout

*Figure 4 continued on next page*

*Figure 4 continued*

mice (*Opn4^{Cre/Cre}*) expressing the $G_q$-coupled DREADD (hM3D) specifically in ipRGCs (AAV2-hSyn-DIO-hM3D($G_q$)-mCherry). CNO injection IP (blue) caused robust constriction within 5–10 min that was sustained for 60 min, whereas PBS injection (black) did not. CNO data is fit with a one-phase association curve and PBS data is fit with a linear regression (*n* = 6, mean ± SD). (E) (top) Diagram showing viral eye injection in only one eye. (bottom) Confocal microscope images of an *Opn4^{Cre/Cre}* retina injected with AAV2-CMV-DIO-mRuby-P2A-Melanopsin-FLAG showing infection and expression (mRuby, top; anti-OPN4, bottom). Scale bar = 50 μm. (F) Successful rescue of pupil constriction by virally restored melanopsin expression in a single eye of adult mice (WT *n* = 6, Mel. KO *n* = 12, Mel.-Rescue *n* = 4). (right) Representative pupil images of Mel. KO and Mel.-Rescue mice at 1000 lux. (G) PLR intensity-response curves of Wildtype (*n* = 6), Mel.-only (Rod-DTA; Cone-DTA *n* = 8), Cone KO (*Cnga3^{-/-}* *n* = 4), and Rod KO (Rod-DTA *n* = 5) mice (mean ± SD). Melanopsin is sufficient at high light (≥1000 lux), however, rods are required at lower light intensities. Cone KO mice are similar to wildtype. (top) Representative pupil images at 1000 lux. See also *Figure 4—figure supplement 1–4*.

The following figure supplements are available for figure 4:

**Figure supplement 1.** Melanopsin is required for sustained constriction across the day.

**Figure supplement 2.** Viral infection and expression is specific to ipRGCs.

**Figure supplement 3.** Rods, but not cones, contribute to sustained PLR sensitivity.

**Figure supplement 4.** Rods drive the residual sustained pupil constriction observed in the absence of melanopsin.

## PACAP is essential for the sustained PLR

Studies of ipRGC neurotransmitters, in combination with our transient PLR results presented here, suggest that glutamate is the primary ipRGC neurotransmitter, and that PACAP plays a minor, or modulatory, role (*Beaulé et al., 2009*; *Colwell et al., 2004*; *Delwig et al., 2013*; *Gompf et al., 2015*; *Kawaguchi et al., 2010*, *2003*; *Purrier et al., 2014*). However, when we tested the sustained PLR in ipRGC glutamate knockout mice, we found that their pupil constriction improved over time compared to their transient PLR sensitivity (*Figure 5B,C*). In contrast, PLR sensitivity either stays the same or declines in all other mutant lines, suggesting that the remaining signal in glutamate knockout mice, presumably PACAP, becomes more effective with longer stimulus duration. Intriguingly, ipRGC glutamate knockout mice showed pulsatile or periodic pupil constriction over time, potentially due to waves of neuropeptide vesicle delivery and release from ipRGC axons (*Video 1*).

Neuropeptides have been shown to require high frequency neuronal activity for release and have relatively slow signaling kinetics compared to classical neurotransmitters (*Vaaga et al., 2014*), suggesting that PACAP may be involved in sustained ipRGC signaling at bright light intensities. In support of a role for PACAP in sustained PLR signaling, we find that even though ipRGC PACAP knockout mice show normal transient PLR, they have an attenuated sustained PLR (*Figure 5B–E*). This deficit in ipRGC PACAP knockout mice occurs even at moderate light intensities (10 and 100 lux). ipRGC PACAP KO mice display decaying constriction over time at 1000 lux as opposed to maintained constriction in wildtype mice and enhanced constriction in ipRGC glutamate KOs (*Figure 5D*). At the brightest light intensity tested, 5000 lux, ipRGC PACAP KO mice display significantly worse sustained constriction than ipRGC glutamate KO mice (*Figure 5E*), suggesting that PACAP is more important than glutamate for maintained responses under daylight conditions (1000–100,000+ lux).

Additionally, we observed similar yet more pronounced deficits in full body PACAP KO mice (*Adcyap1^{-/-}*; *Figure 5—figure supplement 1*). They display wildtype transient responses (*Figure 5—figure supplement 1A,B*) and severely attenuated sustained responses (*Figure 5—figure supplement 1C–E*). Interestingly, these PACAP knockout mice exhibit PLR decay on a similar timescale as melanopsin knockout mice (half-life: ~5 min, *Figure 4C* and *Figure 5—figure supplement 1F*). These results provide evidence that PACAP allows ipRGCs to communicate sustained input to downstream neurons. As observed with the photoreceptor contributions, the highest sensitivity of sustained PLR requires PACAP/glutamate synergy.

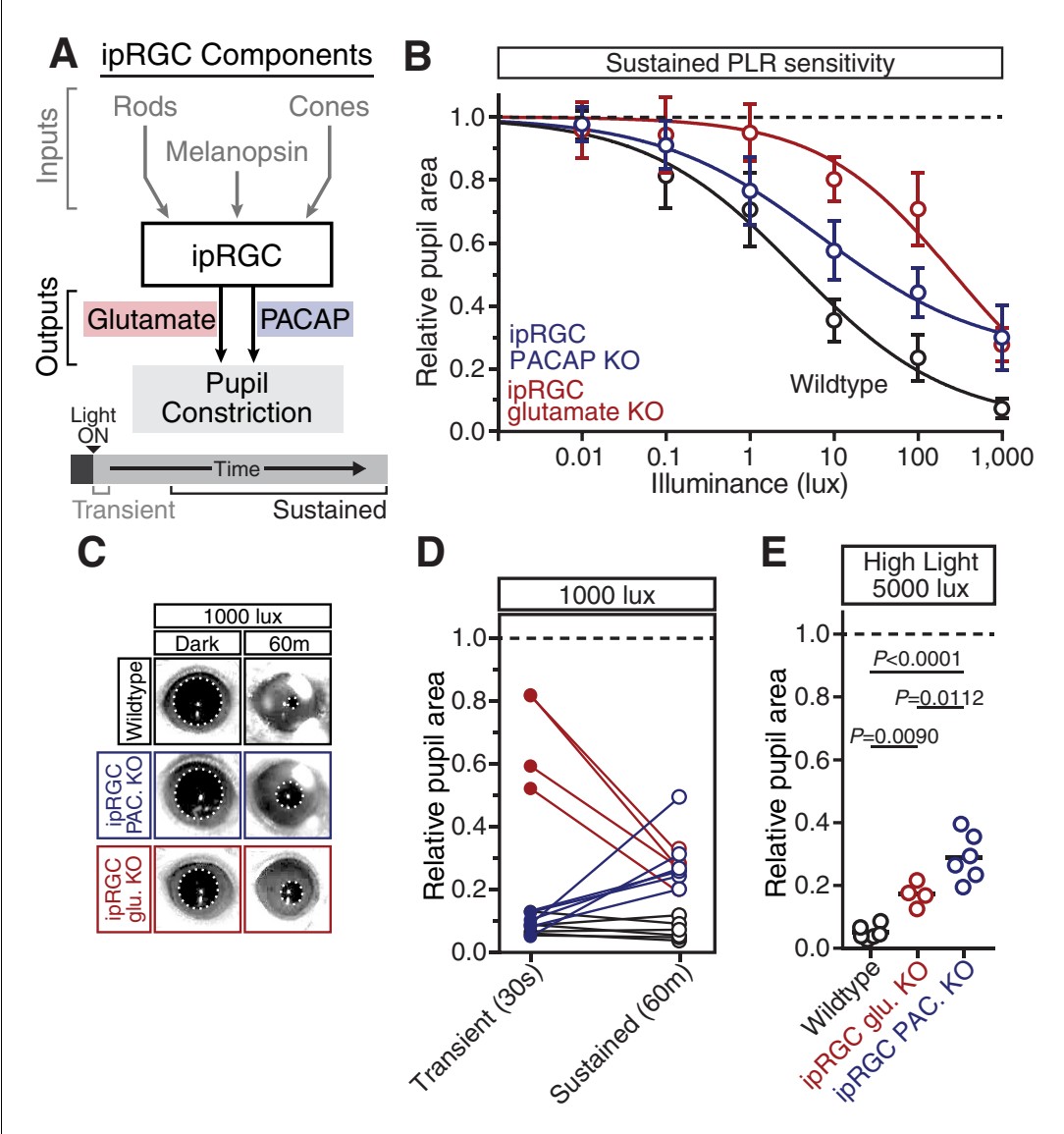

**Figure 5.** PACAP is essential for the sustained PLR. (**A**) Diagram of ipRGC behavioral circuit. (**B**) PLR intensity-response curves of sustained constriction in mice lacking glutamatergic or PACAPergic neurotransmission (WT *n* = 6, ipRGC glu. KO *n* = 4, ipRGC PACAP KO *n* = 6)(mean ± SD). Both mutants display deficits at 10, 100, and 1000 lux as compared to wildtype (wildtype v. ipRGC Glu. KO: 10 and 100 lux p<0.0001, 1000 lux p=0.0004 by two-way ANOVA with Sidak's post-test)(wildtype v. ipRGC PACAP KO: 10, 100, and 1000 lux p<0.0001 by two-way ANOVA with Sidak's post-test). (**C**) Representative pupil images of sustained constriction at 1000 lux. (**D**) Comparison of transient and sustained constriction under high light (1000 lux). ipRGC glu. KO mice (red) show an increase in pupil constriction with time whereas ipRGC PACAP KOs (blue) display a significant loss of constriction over time (ipRGC glu. KO transient v. sustained p<0.0001, ipRGC PACAP KO transient v. sustained p=0.0003, wildtype transient v. sustained p=0.9921 by one-way ANOVA with Sidak's post-test). (**E**) Pupil constriction of neurotransmitter mutant mice after sustained 5000 lux light. Data from individual mice shown with mean (black bar). Statistical significance determined by one-way ANOVA with Sidak's post-test. See also *Figure 5—figure supplement 1*.

The following figure supplement is available for figure 5:

**Figure supplement 1.** PACAP KO mice display similar PLR phenotypes to ipRGC-specific PACAP KO mice.

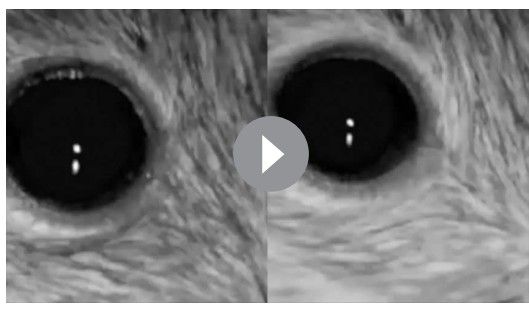

**Video 1.** Pulsatile pupil constriction in the absence of glutamatergic neurotransmission. This video is at 5x speed. 1000 lux white light (6500K) turns on at approximately 1s.

## Model of ipRGC circuit transitions

Based on our results, we generated a quantitative representation of the distinct roles played by each photoreceptor input and neurotransmitter output of ipRGCs for the PLR over a range of light intensities and light stimulus durations (*Figure 6*, see Materials and methods for detailed explanation). We integrated individual necessity (i.e. from knockout lines) and sufficiency (i.e. from '−only' lines) of rods, cones, and melanopsin in driving the PLR (*Figure 6—figure supplement 1*) to generate a merged heat map representing each photoreceptor's input to the PLR (*Figure 6A,B*). We then performed the same technique to represent the neurotransmitter outputs of ipRGCs for the PLR (*Figure 6C,D* and *Figure 6—figure supplement 1*) using only the necessity heat maps because we cannot rule out the possibility that other neurotransmitters contribute to ipRGC function. These heat maps provide a comprehensive visualization of the contribution made by each photoreceptor's input and each neurotransmitter's output for ipRGC signaling at any particular time or environmental light intensity. ipRGC transient signaling for the PLR is dominated by input from rods (*Figure 6A*, red) and output by glutamate (*Figure 6C*, green). In contrast, sustained PLR signaling is dominated by melanopsin (*Figure 6B*, blue) and PACAP (*Figure 6D*, blue). Together, these experiments and our model highlight a mechanistic transition in the ipRGC circuit supporting transient and sustained behavioral outputs.

## Discussion

We show here how inputs and outputs for a specific circuit change across time to support a behavioral response. Remarkably, the mechanisms supporting transient and sustained responses are distinct, suggesting stimulus duration as a critical determinant of circuit state. Transient PLR responses predominantly utilize classical, well-characterized visual system synaptic mechanisms: rod phototransduction and signal relay to ipRGCs, followed by ipRGC glutamatergic output. However, as conventional signaling mechanisms adapt, non-conventional mechanisms are recruited to maintain persistent signaling, including endogenous melanopsin phototransduction and peptidergic neurotransmission through PACAP. Our findings highlight fundamental circuit changes in the light-adapted retina that are relatively unexplored (*Tikidji-Hamburyan et al., 2015*).

Our results reveal the roles of distinct photoreceptors and neurotransmitters in the PLR and probably other ipRGC-dependent behaviors. We show how ipRGC inputs and outputs can contribute to the PLR through changes in their relative contribution across stimulus intensity and duration. Our ability to decipher these elaborate dynamic changes stems from the fact that we used a large array of environmental light intensities and durations, coupled with genetic means to silence individual circuit components. Ultimately, our quantitative model makes testable predictions about the role of each photoreceptor and neurotransmitter for other ipRGC-dependent behaviors.

We show that in contrast to many proposed models, rods provide the exclusive transient input to ipRGCs for the PLR at dim (scotopic) and moderate (mesopic) light intensities. That rods are capable of rapid and sensitive input to ipRGCs is not surprising given electrophysiological evidence of sensitive rod input to ipRGCs (*Weng et al., 2013*; *Zhao et al., 2014*) and the fact that rods are widely appreciated as the mediators of dim light vision. However, their exclusive input at mesopic light intensities suggests that cone input to ipRGCs is relatively weak, consistent with the inability of cones to drive circadian photoentrainment (*Lall et al., 2010*; *Mrosovsky and Hattar, 2005*). Furthermore, we report here that in addition to their role in high-sensitivity transient signaling, rods are capable of driving sustained signaling at bright light intensities well above their saturation level (~40 lux, *Figure 4—figure supplement 4*). This agrees with previous findings that rods are capable of supporting circadian photoentrainment at bright light intensities (*Altimus et al., 2010*) but also

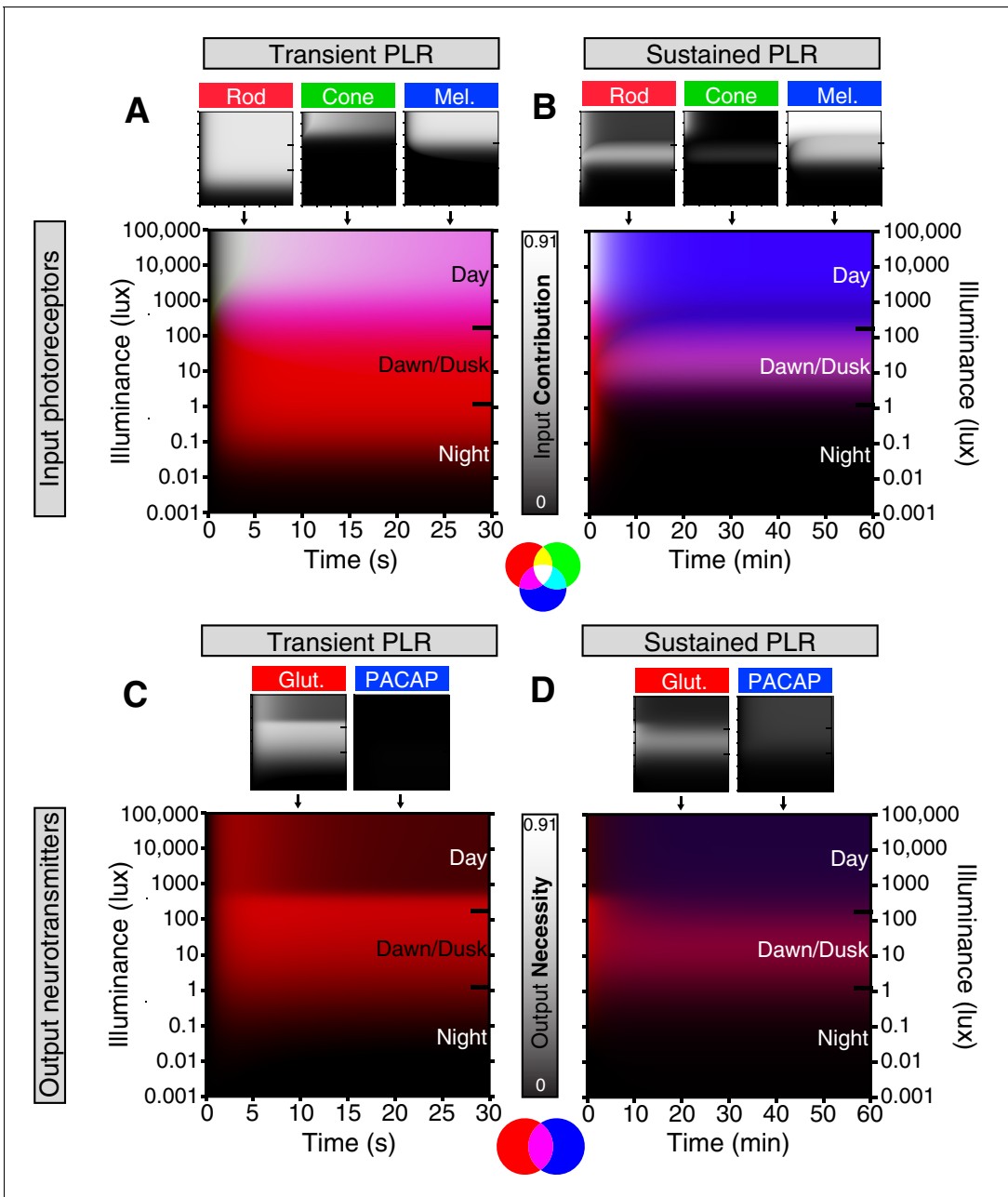

**Figure 6.** Model of ipRGC circuit transitions. (A and B) Heat maps of (A) transient and (B) sustained PLR as duration and intensity vary. Night, dawn/dusk, and daytime light intensities indicated by ticks on right side of plot. (top) Heat maps of individual photoreceptor contributions (grayscale). Black represents no contribution and degree of white represents increasing contribution. Each photoreceptor contribution heat map is a combination of necessity (individual photoreceptor transduction knockouts) and sufficiency ('photoreceptor-only') heat maps (for example: Input Contribution$_{rod}$ = Max (Necessity$_{rod}$, Sufficiency$_{rod}$)). (middle) Rod (red), cone (green), melanopsin (blue) contributions are combined into a single heat map. (bottom) Color combination guide for reference when viewing heat map. (C and D) Same as above for neurotransmitter contributions to transient (C) and sustained (D) ipRGC signaling. Glutamatergic contribution is in green and PACAPergic contribution is in blue. See the Materials and methods section for details on heat map generation. Note that the axes are the same for the individual and combined heatmaps. See also *Figure 6—figure supplement 1*.

The following figure supplement is available for figure 6:

**Figure supplement 1.** Necessity/Sufficiency heat maps for photoreceptor input to pupil constriction.

provides more precise temporal kinetics of rod input to subconscious behaviors. It has been proposed that rods never fully saturate (*Blakemore and Rushton, 1965*), and here we provide a physiological role for rod activity at daylight intensities.

In contrast to previous data that melanopsin is largely dispensable for the PLR (*Lucas et al., 2003*), we find that it is the dominant determinant of pupil size during the day. This is likely due to the fact that rod and cone inputs adapt to background light, while we find no evidence of behavioral light adaptation in melanopsin phototransduction (i.e. identical sensitivity of melanopsin-only mice in transient and sustained PLR). While melanopsin phototransduction adapts in vitro (*Do and Yau, 2013*; *Wong et al., 2005*), it has been proposed that only the adapted state is able to influence downstream behaviors (*Do and Yau, 2013*). We predict that melanopsin will be required in other visual functions throughout the day, for example as in more natural photoentrainment conditions that need to precisely measure changing light intensity under bright conditions or measuring day length (*Gooley et al., 2010*; *Mrosovsky and Hattar, 2003*; *VanderLeest et al., 2007*). This requirement for melanopsin in sustained light detection is likely the main reason melanopsin has been conserved in vertebrates.

To date, glutamatergic neurotransmission is the only retina-brain signaling mechanism that has been robustly characterized. We confirm previous data that ipRGCs predominantly rely on glutamatergic output for the transient PLR (*Delwig et al., 2013*; *Gompf et al., 2015*; *Purrier et al., 2014*). However, we show that the stimulus durations in which glutamate predominates over PACAP is relatively restricted (<5 min), revealing the first critical role for a neuropeptide in retinal signaling to the brain. Further, we find that PACAP appears sufficient to drive the PLR independent of its potential to modulate glutamate. There have been discrepancies in the literature about the role of PACAP in the PLR (*Engelund et al., 2012*; *Kawaguchi et al., 2010*), which we believe is likely due to differences in light stimulus duration. Intriguingly, PACAPergic neurotransmission appears to be pulsatile, potentially reflecting the imprecision of slow vesicle delivery from the soma and suggesting why ipRGCs also require a fast and reliable glutamatergic signal. Glutamate and PACAP are the only known ipRGC neurotransmitters, but it remains possible there are neurotransmitters which remain undiscovered. An ipRGC-specific glutamate/PACAP double knockout is a crucial next step in understanding ipRGC neurotransmission. Given the expression of other neuropeptides in many RGCs, including ipRGCs (*Brecha et al., 1987*; *Djeridane, 1994*; *Kay et al., 2011*; *Liu et al., 2011*), it remains possible that neuropeptides have a broader role in visual function than previously appreciated.

The complementary arrangement of inputs and outputs for the PLR we describe here demonstrates how the visual system accomplishes high sensitivity, transient responses as well as integrative, long-term responses. Many other signaling systems may employ discrete methods for signaling robustly through time. While melanopsin is specific to the ipRGC circuit, PACAP and other neuropeptides may play similar roles in long-term signaling in other circuits, such as hypothalamic feeding circuits (*Krashes et al., 2013*). Expanding the timescales over which we investigate these systems is likely to reveal entirely new aspects of cell signaling.

## Materials and methods

### Animal husbandry

C57Bl/6 × Sv129 hybrid mice were used in all experiments except PACAP KO mice which were C57Bl/6. All mice were housed according to guidelines from the Animal Care and Use Committee of Johns Hopkins University. Male and female mice age 2–8 months were housed in plastic translucent cages with steel-lined lids in an open room. Ambient room temperature and humidity were monitored daily and tightly controlled. Food and water were available *ad libitum.* All mice were maintained in a 12 hr:12 hr light-dark cycle with light intensity around 100 lux for the entirety of their lives.

### Pupillometry

All mice were dark-adapted for at least 30 min prior to any experiments and all PLR experiments were performed between Zeitgeber times (ZT) 2 and 10. For all experiments, mice were unanesthetized and restrained by hand. Because stress can affect pupil size, we ensured that the mice were

not stressed during these experiments. To do so, we handled the mice for several days prior to the experiments to get them accustomed to the researchers and to being scruffed. Any mice that showed signs of stress, including vocalizations and wriggling during the experiments, were not used and were subjected to more handling sessions before use in experiments.

Mice were restrained manually under a 10-, 13-, or 23-Watt compact fluorescent light bulb (GE Daylight FLE10HT3/2/D or Sylvania Daylight CF13EL and CF23EL) with a color temperature of 6500 K to simulate natural sunlight. The light intensity was measured using a light meter (EXTECH Foot Candle/Lux Light Meter, 401025) at the surface on which the mouse was held. The light meter was initially calibrated by EXTECH using a Tungsten 2856 K light source; because our experiments used a fluorescent bulb of 6500 K, all measured light intensities reported here may vary by 0.92–1.12 times the actual light intensity. Light intensity was adjusted by a combination of altering the distance of the light bulb(s) from the mouse and/or applying neutral density filters (Roscolux). The light meter is incapable of detecting light intensities below 1 lux, so one neutral density filter cutting the light intensity by 12.5% was applied to the bulb to estimate 1-log unit decreases in illumination below 1 lux. Light intensities above 500 lux required the use of multiple light bulbs.

For the monochromatic light PLR experiments, an LED light (SuperBrightLEDs) was housed in a microscope light source with fiber optic gooseneck arms to direct the light source to the mouse eye. For the experiments involving the $Opn1mw^{red}$ mice, we used a 626-nm LED in this setup and directed light to both eyes simultaneously or to just one eye and measured the PLR in the illuminated eye (see figure legends). The photon flux was measured using a luminometer (SolarLight) and converted from $W/m^2$ to $photons/cm^2/s$. The light intensity was decreased by 12.5% using neutral density filters (Rosco).

Videos of the eye were taken using a Sony Handycam (DCR-HC96) mounted on a tripod a fixed distance from the mouse. Manual focus was maintained on the camera to ensure that only one focal plane existed for each mouse and that therefore variable distance from the camera should not contribute to differences in relative pupil area throughout the video. Pupil size was first recorded under dim red light and the endogenous infrared light source of the camera to capture the dark-adapted pupil size. Following at least 5 s of recording in dark, the pupil was continuously recorded for at least 30 s of a light step stimulus. For all sustained PLR, animals were kept in a cage for 60 min under the light stimulus. Animals were removed from the cage after 60 min and held in front of the camera for 30 s as for the transient PLR. All pupil images presented in the paper were cropped to a fixed square area (generally 100 × 100 pixels) surrounding the eye using GNU Image Manipulation Program (GIMP). The images were made grayscale and then brightness and contrast were adjusted to enhance visibility of the pupil and exported as PNG files.

## Data analysis

Videos were transferred from the camera to a computer as Audio Video Interleave (AVI) files and individual frames were taken using VLC media player (www.videolan.org/vlc/) and saved in portable network graphics format (PNG). Images were taken in the dark, at 5 s, and 30 s following stimulus onset. Pupil area was then quantified manually in ImageJ (http://rsbweb.nih.gov/ij/) software. The pupil area was measured in pixels using the oval tool in which the 4 cardinal points of the oval were touching their respective edges of the pupil. The relative pupil area was calculated using LibreOffice Calc or Microsoft Excel in which the area during the light stimulus was divided by the area prior to lights onset. For the transient PLR, the minimum relative pupil size of either 5 s or 30 s after stimulus was used for all genotypes.

The intensity-response curve was fit using a variable slope sigmoidal dose-response curve in Graphpad Prism 6. The top and bottom of the fit were constrained to 1.0 and between 0 and 0.10, respectively, to ensure the $EC_{50}$ for each genotype was represented by similar curves. For genotypes that never showed evidence of reaching between 0 and 0.10 relative pupil size, the bottom was not constrained. The sensitivity for each genotype was calculated using the same process of fitting each individual animal's data points with a sigmoidal dose-response curve to generate $EC_{50}$.

## Conditional PACAP allele

The lox-modified *PACAP (Adcyap1)* targeting construct was made by recombineering technology. To engineer the targeting vector, 5' homology arm, 3' homology arm and CKO region were

amplified from mouse Sv129 BAC genomic DNA and confirmed by end sequencing (Cyagen biosciences, Santa Clara, CA). The two *loxP* sites flank the second exon and when recombined, create a frameshift mutation and truncated protein. The plasmid was electroporated into W4 ES cells and cells expanded from targeted ES clones were injected into C57BL6 blastocysts. Germline transmitting chimeric animals were obtained and then mated with flpE mice to delete the *frt*-site flanked neomycin selection cassette. The resulting heterozygous offspring were crossed to generate homozygous PACAP^lox/lox study subjects. All mice are thus on a mixed C57Bl6/J and 129Sv background. Offspring were genotyped by PCR using 2 primers (F: CCGATTGATTGACTACAGGCTCC and R: G TGTTAAACACCAGTTAGCCACGC) which detect the presence or absence of the 5′ loxP site and a 3rd primer was used in conjunction with the forward primer (CKO-R GGGCTTTGATCTGGGAAC TGAAG) to detect the recombination event. By generating mice homozygous for a germline deleted cre-deleted allele, we have established that the cre-deleted allele does not express intact *PACAP* mRNA (by PCR and by ISH). A more detailed description of the generation and use of the allele will appear in a manuscript that is in preparation (Ross and Lowell, unpublished).

## Viral infection

Mice were anesthetized by intraperitoneal injection of avertin (2, 2, 2-Tribromoethanol) and placed under a stereo microscope. 1 µl of AAV2-hSyn-DIO-hM3DG$_q$-mCherry ($4.6 \times 10^{12}$ viral particles/ml, Roth lab, UNC Vector Core) or AAV2-CMV-DIO-mRuby-P2A-Melanopsin-FLAG (Robinson lab, UMBC) was placed on a piece of Parafilm and drawn into a 10-µl microcapillary tube (Sigma P0674) that had been pulled to a needle (Sutter Instruments, Model P-2000). The loaded needle was then placed in the holster of a pico-injector (Harvard Apparatus PLI-90). The needle punctured the eye posterior to the ora serrata and air pressure was used to drive the viral solution into the vitreous chamber of the eye to ensure delivery specifically to the retina. Mice recovered from surgery on a heating pad until they woke from anesthesia. All PLR experiments and confocal imaging were done at least 3 weeks following viral injection.

## Immunofluorescence and confocal microscopy

Mice that had been infected with the AAVs were anesthetized with avertin and then euthanized using cervical dislocation. The eyes were removed and the retinas were dissected in PBS and then fixed in 4% paraformaldehyde for 1–2 hr on ice. The retinas were then washed in PBS at least three times before mounting on a microscope slide (Fisher, Hampton, NH) in Fluoromount (Sigma, St. Louis, MO) with DAPI (2-(4-amidinophenyl)-1H -indole-6-carboxamidine). Some retinas were co-stained for melanopsin using rabbit anti-OPN4 (Advanced Targeting Systems, San Diego, CA, AB-N38, 1:1000) in 4% goat serum with secondary antibody Alexa Fluor 488 goat anti-rabbit (Life Technologies, Carlsbad, CA, A11008, 1:1000). Images were taken on a Zeiss LSM 710 confocal microscope using a 20× objective. After imaging, images were made grayscale, background subtracted, and brightness and contrast were adjusted in FIJI (http://fiji.sc) for the image presented in the paper.

## Statistical analysis

All statistical tests were performed in Graphpad Prism 6. Specific statistical comparisons are listed in the figure captions. Because the $EC_{50}$ data appears to be a normal distribution on a log scale (lognormal distribution), all statistical tests and data analysis involving $EC_{50}$ were performed on the log transformed data set.

## Heat map generation

The photoreceptor contribution heat map was generated by first creating estimated pupil size matrices for the both the rapid and sustained PLR at every light intensity and time for wildtype mice (x axis = time, y axis = intensity). To do so, we applied the equation for a one-phase association:

$$Y = Y0 + (Plataeu - Y0) * \left(1 - e^{(-K*x)}\right)$$

In our case, $Y$ is the relative pupil area generated at time, $x$. For the WT rapid PLR heat map, $Y0_{rapid}$ is set to 1 for every light intensity and the $K_{rapid}$ was extracted from the wildtype rapid constriction

kinetics curve at 100 lux. The $Plateau_{rapid}$ value at each light intensity is the rapid PLR value extracted from the WT rapid intensity-response curve fit. This allows us to generate a full matrix of WT pupil sizes at every intensity and time by knowing the final pupil size ($Plateau$) and the rate of constriction ($K$). This then generates a full matrix of values for every time and intensity for WT mice.

The same method was applied to make the sustained PLR heat map. However, in this case, $Y0_{sustained}$ was set to the value of the rapid PLR at each light intensity (e.g. the same value as $Plateau_{rapid}$). The $Plateau_{sustained}$ value is extracted from the sustained intensity-response curve fit at each intensity. The $K_{sustained}$ was extracted from our wildtype sustained time courses (*Figure 1c*). Because the decay rate for sustained constriction appeared to change with intensity (*Figure 1f*) we used a sigmoidal curve fit to our experimentally determined decay rates (1, 10, 100 lux) to generate decay rates for a range of light intensities. We constrained the top and bottom of this curve to the decay rates determined for 1 and 100 lux respectively.

This process was used to generate two matrices of relative pupil areas with the y-axis being light intensity varying logarithmically (0.001–100,000 lux) and the x-axis being time varying linearly from 0 to 30 s for the rapid and 30 s to 60 min for the sustained. This was done using a custom MATLAB script.

The matrices generated for the wildtype mice were also done to the photoreceptor mutants. In order to determine necessity of a photoreceptor we subtracted rod (average of $Gnat1^{-/-}$ and Rod-DTA), cone (average of $Cnga3^{-/-}$, $Gnat2^{-/-}$ and Cone-DTA), or melanopsin ($Opn4^{-/-}$) knockout matrices from the wildtype matrix. This yields larger values for genotypes that are more required and also normalizes for the overall constriction in wildtype mice at that intensity (i.e. because rods are fully necessary at some dim intensities at which WT mice have minimal constriction, the necessity value attributed to rods is small despite their absolute necessity at that intensity). To determine sufficiency we used 'rod-only' ($Cnga3^{-/-}$; $Opn4^{-/-}$), 'cone-only' ($Gnat1^{-/-}$;$Opn4^{-/-}$) and 'melanopsin-only' (average of $Gnat1^{-/-}$;$Gnat2^{-/-}$, $Gnat1^{-/-}$; $Cnga3^{-/-}$ and Rod-DTA;Cone-DTA) matrices. Additionally, we applied the decay rate of pupil constriction from the 'cone-only' mouse line transient PLR at 100 lux for all light intensities.

Finally, matrices generated above were exported as heat map images with MATLAB.

## Negative feedback modeling

In order to isolate negative feedback's impact on the PLR, we generated a computational model. Computational modeling was performed with MATLAB using two experimentally determined parameters. First, the relative pupil area (RPA) values for the wildtype intensity-response curve (*Figure 1d*). These values give us the response driven when the pupil starts fully open. We will later multiply the environmental intensity by the new relative pupil area to determine the new retinal intensity. We will use this new retinal intensity to extract the pupil size from the rapid intensity-response curve to find the constriction driven by that new intensity under baseline conditions. The model does this recalculation of retinal intensity and the PLR driven by it every second for 956 s.

The second experiment integrated into the model is a 1 s light pulse-chase experiment. Here, we dark-adapted the mouse, gave a single second of light and then followed subsequent constriction for 30 s. These constriction values were normalized to the maximum constriction achieved, in this case the 6-s time point. This gives us the ability to weight the contribution of light at a particular time to constriction at subsequent times. As you can see, light does not instantly constrict the pupil. It takes several seconds for the signal to maximally impact pupil size, which is then followed by signal decay. Importantly, this temporal weighting, while not required for the model, does give us a rough estimate of the potential kinetics of feedback's impact on PLR decay.

With these pieces of experimental data in hand, the model does the following at every light intensity (0.0001–100,000 lux): (1) it extracts the RPA in response to a particular light intensity from the wildtype intensity-response curve. (2) The model uses the temporal weighting values from the pulse-chase experiment to weight that RPA across subsequent times (0–30 s). This gives us a 30-s constriction time course for the light detected at time zero. (3) The model next moves to time 1 s. Now it takes into account the maximum constriction caused by light at previous times (time 0 in this case). The model uses that constriction to reduce the light intensity and calculate a new retinal light intensity: RPA *Light intensity = Retinal intensity. (4) Next, it determines the RPA driven by this new retinal intensity using the DRC once again. (5) Repeats step (2) for this RPA giving another time course of constriction (1–31 s). (6) The model repeats steps (3–5) moving up in 1s increments each time.

Importantly, at each new time point it finds the maximum constriction value in response to all previous time points in order to calculate the new retinal intensity. (7) Finally, it finds the maximum constriction at each time point in order to produce a negative feedback PLR decay time course. See graphical representation of the negative feedback model (*Figure 1—figure supplement 2A*)

*The primary assumption the model makes is that the PLR system has zero summation of signal. This is probably unlikely. However, this assumption was made to maximize the impact of feedback on pupil constriction. This model provides us with an upper bound on negative feedback's contribution to PLR decay.

*Source code and materials used are available on Github (https://github.com/keenanw27/PLR-Decay-Model).

## Mathematical description of the negative-feedback model of PLR decay

At a given environmental light intensity: $lux_o$. The effect of pupillary negative-feedback during a 956s stimulation is modeled as follows:

$$for\ time\ t = 1,\ 2,\ 3\ldots 956$$
$$max\left(\overrightarrow{RPA}(:,t)\right) \times lux_o = lux_t \tag{1}$$

In equation (1) above, we determine the retinal light intensity, $lux_t$, that is, the intensity of light after modulation by pupil size at time $t$. At $t = 1$ there is no pupil constriction and therefore no light intensity modulation ($lux_o = lux_t$). $\overrightarrow{RPA}$ is a $956 \times 956$ matrix which stores subsequent pupil constriction values. With $lux_t$ we determine the constriction driven by light sensed at time, $t$:

$$\overrightarrow{\alpha}(lux_t) \times \overrightarrow{\omega} = \overrightarrow{RPA}(t,t{:}t+30) \tag{2}$$

In equation (2), we calculate the amount of constriction driven by $lux_t$, $\overrightarrow{\alpha}(lux_t)$, and approximate the temporal characteristics of that constriction with $\overrightarrow{\omega}$. $\overrightarrow{\omega}$ is based on a 1s light pulse-chase experiment where we followed the constriction driven by 1 s of light for 30 s. Again, we store calculated constriction values: $\overrightarrow{RPA}(t,t{:}t+30)$. Finally, we extract the highest constriction value at $t$:

$$max(\overrightarrow{RPA}(:,t)) = \overrightarrow{Model}_{lux_o}(1,t) \tag{3}$$

After completing $t = 956$, $\overrightarrow{Model}_{lux_o}$ is a vector containing the model-predicted timecourse of pupil constriction when negative-feedback is the only source of PLR decay.

## Acknowledgements

We would like to thank Rejji Kuruvilla, Haiqing Zhao, and other members of the JHU Mouse Tri-Lab for their helpful comments on this project in general and the manuscript in particular. We would like to thank Alex Kolodkin for comments on an earlier version of the manuscript. We would also like to thank Lee E Eiden for providing PACAP KO mice in addition to experimental insight and comments on the manuscript. This work was supported by National Institutes of Health grants GM076430 and EY024452.

## Additional information

### Funding

| Funder | Grant reference number | Author |
| --- | --- | --- |
| National Eye Institute | R21 EY024452 | William Thomas Keenan<br>Alan C Rupp<br>Samer S Hattar |
| National Institute of General Medical Sciences | RO1 GM076430 | William Thomas Keenan<br>Alan C Rupp<br>Samer S Hattar |

| National Eye Institute Intra-mural research program | EY000504-06 | Tudor C Badea |
| National Heart Lung and Blood Institute | 5T32HL007374-36 | Rachel A Ross |
| Harvard Medical School De-partment of Psychiatry | Dupont Warren Fellowship | Rachel A Ross |

The funders had no role in study design, data collection and interpretation, or the decision to submit the work for publication.

## Author contributions

WTK, ACR, Conception and design, Acquisition of data, Analysis and interpretation of data, Drafting or revising the article, Contributed unpublished essential data or reagents; RAR, SSH, Conception and design, Analysis and interpretation of data, Drafting or revising the article, Contributed unpublished essential data or reagents; PS, SH, ZW, TCB, PRR, BBL, Conception and design, Drafting or revising the article, Contributed unpublished essential data or reagents

## Author ORCIDs

William Thomas Keenan, http://orcid.org/0000-0003-3381-744X
Alan C Rupp, http://orcid.org/0000-0001-5363-4494
Samer S Hattar, http://orcid.org/0000-0002-3124-9525

## Ethics

Animal experimentation: This study was performed in strict accordance with the recommendations in the Guide for the Care and Use of Laboratory Animals of the National Institutes of Health. All mice were housed according to guidelines from the Animal Care and Use Committee of Johns Hopkins University (Protocol # MO16A212), and used protocols approved by the JHU animal care and use committee.

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
