## [Decision Letter]

Thank you for submitting your article "A visual circuit uses complementary mechanisms to support transient and sustained pupil constriction" for consideration by *eLife*. Your article has been reviewed by two peer reviewers, and the evaluation has been overseen by a Reviewing Editor (Constance Cepko) and a Senior Editor.

The reviewers have discussed the reviews with one another and the Reviewing Editor has drafted this decision to help you prepare a revised submission.

Summary:

The photoreceptor inputs that regulate the different types of pupillary responses (transient and sustained) were examined in mice using genetic and chemogenetic methods. In addition, the roles played by intrinsically photosensitive ganglion cells (ipRGCs), as well as the roles of glutamate and PACAP, were examined. Although some aspects of these questions have been examined in previous studies, the quality of the work presented in this study exceeds that of previous publications, and extends the work by virtue of the methods and strains used.

Essential revisions:

The demonstration of PACAP as a mediator of the sustained response is ascribed to the ipRGCs. However, since this peptide is expressed more broadly than in ipRGCS, and since a complete ko was used, it was not established that the role of this peptide in pupillary response is due to its release from ipRGCs. This will need to be addressed, most definitively through the use of an ipRGC-specific loss of function. This may require acquisition and breeding of additional mouse strains. If this is the case, the authors can extend the time for revision to accomplish this experiment.

*Reviewer #1:*

In this manuscript, Keenan et al. set out to isolate which photoreceptor inputs and neurotransmitter outputs of intrinsically photosensitive retinal ganglion cells (ipRGCs) mediate the transient and sustained phases of pupil constriction. The authors examine changes in the pupillary light response (PLR) of unanaesthetized mice over time with a novel setup utilizing an overhead, broad-spectrum light source to simulate a range of natural sunlight conditions. Multiple mutant mouse lines are used to identify the photoreceptor inputs and neurotransmitter outputs that contribute to the rapid and stable responses of ipRGCs. Their results demonstrate that distinct mechanisms are involved in the transient and sustained pupil responses: rod phototransduction and glutamatergic output of ipRGCs predominately mediate fast, transient pupil constriction whereas endogenous melanopsin phototransduction and peptidergic (PACAP) output take over to maintain a steady state of pupil constriction.

This study fills in an important gap in knowledge about the behavioral contributions of photoreceptor inputs to ipRGCs and their neurotransmitter outputs. The manuscript is well written and the work appears to be carefully done, although I do have some concern about the wide range of mouse ages used and whether any of the observed differences could have been age-dependent. The experiments to rescue ipRGC function by chemogenetic activation and melanopsin restoration were very nicely done and will be useful tools for future studies.

The authors state throughout the paper that, depending on the duration of the light stimulus (transient vs. sustained), the combination of inputs/outputs of ipRGCs mediating pupil constriction change. However, the input/outputs of ipRGCs seem to also depend on wavelength(s) and intensity of the light stimulus. This is mentioned in the manuscript, but should maybe be discussed further. For example, rod phototransduction and glutamatergic neurotransmission of ipRGCs predominately mediate the transient pupil response but at high light intensities the transient response seems to be driven by intrinsic melanopsin signaling and peptidergic output of ipRGCs.

Overall an important and valuable contribution to the field.

*Reviewer #2:*

Keenan and colleagues have conducted a systematic analysis of the photoreceptors that drive pupil constriction in the mouse. This has been an area of some controversy since the intrinsically photosensitive retinal ganglion cells (ipRGCs) were discovered almost fifteen years ago, and this work holds promise for settling many of the fundamental issues. The importance of this paper also extends beyond the pupil, to the many other functions that are mediated by these photoreceptors, and connects to the generally elusive topic of peptidergic neurotransmission. There is much to admire here. For example, the authors provide control experiments that will be of great use to the field, marshal quite the array of mutations (some overlapping informatively), and do not shy from difficult measurements. Overall, the experiments are impressive, the figures are clear, and the writing is excellent.

Alterations observed in the pupil reflex are interpreted in terms of ipRGCs and the rod/cone inputs these cells receive (e.g., subsection “pRGC behavioral responses are composed of both transient and sustained phases”, last paragraph). However, are ipRGCs the only RGCs driving the pupil? More specifically, when ipRGCs are ablated, does any retinal innervation of the OPN remain?

PACAP expression is not exclusive to ipRGCs. Therefore, controls are required before the authors can attribute a defect in the global PACAP knockout to the lack of PACAP released from ipRGCs. Rescuing PACAP within ipRGCs would be one way forward. For the Vglut2 knockout, it would be important to demonstrate actual loss of glutamate in ipRGCs and normal glutamate elsewhere (given the documented leakiness of Cre expression in the mouse line used). Some discussion of potential developmental effects would be helpful.

With regard to Figure 2—figure supplement 3, it appears clear that naturalistic illumination can produce maximal, transient pupil constriction without melanopsin. However, is this experiment comparable to the two others that are presented? These others use different light sources. How do their spectra and intensities compare at the retina, in terms of activating the rods, cones, and ipRGCs?

Omission of stimulus details makes it difficult to fully assess whether the impressive number of rod, cone, and ipRGC mutants employed are behaving as expected (or if there are interesting deviations from expectation). The absorption spectra of known mouse pigments should also be considered more rigorously in assessing the strengths of the various stimuli (e.g., for Figure 1—figure supplement 1).

The model of pupil constriction (Figure 1—figure supplement 2) has no theoretical or mechanistic basis and, I think, detracts from the paper. Measurement of the consensual pupil reflex with the illuminated eye dilated or not should suffice. I assume this is the experiment shown in Figure 1—figure supplement 2 (the legend does not state whether only one eye was illuminated). If so, the lack of an effect is surprising because the intensity used, 10 lux, should produce substantial constriction of the illuminated eye. Additional discussion is required. Could this experiment be repeated at different light intensities, for direct comparison to the plots in panel C?

---

## [Author Response]

*Essential revisions:*

*The demonstration of PACAP as a mediator of the sustained response is ascribed to the ipRGCs. However, since this peptide is expressed more broadly than in ipRGCS, and since a complete ko was used, it was not established that the role of this peptide in pupillary response is due to its release from ipRGCs. This will need to be addressed, most definitively through the use of an ipRGC-specific loss of function. This may require acquisition and breeding of additional mouse strains. If this is the case, the authors can extend the time for revision to accomplish this experiment.*

*Reviewer #2:*

*Alterations observed in the pupil reflex are interpreted in terms of ipRGCs and the rod/cone inputs these cells receive (e.g., subsection “pRGC behavioral responses are composed of both transient and sustained phases”, last paragraph). However, are ipRGCs the only RGCs driving the pupil? More specifically, when ipRGCs are ablated, does any retinal innervation of the OPN remain?*

ipRGCs provide virtually all functional input to the PLR. When the majority of M1 ipRGCs are ablated, a small residual PLR remains only at bright light intensities (Guler et al. Nature2008) and when all ipRGCs are ablated, no PLR remains (Hatori et al. Plos One2008).

Morphologically, in Alen et al. Nature2011 paper, eliminating all Brn3b-positive ipRGCs removed all innervations to the OPN.

Together, we believe that ipRGCs are essentially the only input to the PLR.

*PACAP expression is not exclusive to ipRGCs. Therefore, controls are required before the authors can attribute a defect in the global PACAP knockout to the lack of PACAP released from ipRGCs. Rescuing PACAP within ipRGCs would be one way forward.*

We thank the reviewer for this suggestion. To address this concern, we have collaborated with Bradford Lowell’s lab (now 2 new authors on this manuscript) to cross a PACAP- floxed line to our melanopsin Cre line (*Opn4_Cre/_ ; Adcyap1_fl/-_*) to eliminate PACAP from ipRGCs specifically. As in the case of the PACAP Knockout animals, these conditional knockout animals showed deficits in the sustained, but not transient, PLR. We now include this data in the main figure (Figure 3 and Figure 5) and move the PACAP conventional knockout to supplement (Figure 5—figure supplement 1).

*For the Vglut2 knockout, it would be important to demonstrate actual loss of glutamate in ipRGCs and normal glutamate elsewhere (given the documented leakiness of Cre expression in the mouse line used). Some discussion of potential developmental effects would be helpful.*

The ipRGC-Vglut2 KO mouse has been used and characterized in multiple previous papers and the loss of Vglut2, specific to ipRGCs, has been confirmed (Delwig et al. 2013 (Figure 1) and Purrier et al. 2014 (Figure 1—figure supplement 1–Figure 1—figure supplement 2)).

As for developmental effects, the melanopsin expression starts weakly at E15 and occurs maximally postnatally (P0-4). We have also checked the innervation of targets for both ipRGCS and other conventional ganglion cells and they appear normal. So, we believe there are no major developmental deficits in these animals, but we cannot fully eliminate subtle changes at this time. Therefore, we are not sure what to discuss as a potential developmental changes due to the lack of glutamate input from ipRGCs.

*With regard to Figure 2—figure supplement 3, it appears clear that naturalistic illumination can produce maximal, transient pupil constriction without melanopsin. However, is this experiment comparable to the two others that are presented? These others use different light sources. How do their spectra and intensities compare at the retina, in terms of activating the rods, cones, and ipRGCs?*

First, we apologize for the oversight in omitting the light intensities from the figure legend. We now include this data in the revised manuscript. The light intensities for the 3 conditions are: Blue light contralateral: 1.9×10_16_ photons/cm_2_/s (very bright), White light contralateral: 27.58 W/m_2_, White light overhead: 4.4 W/m_2_.

Therefore, we believe the difference between contralateral and overhead is not due to the spectrum of the light since white light was much of higher intensity in the contralateral than the overhead (27.58 compared to 4.4) situation, but still we had better pupil constriction under overhead when compared to contralateral. We honestly have no explanation for this result. All we could do is speculate that integrating light input from both eyes somehow can compensate for the absence of melanopsin from ipRGCs.

*Omission of stimulus details makes it difficult to fully assess whether the impressive number of rod, cone, and ipRGC mutants employed are behaving as expected (or if there are interesting deviations from expectation). The absorption spectra of known mouse pigments should also be considered more rigorously in assessing the strengths of the various stimuli (e.g., for Figure 1—figure supplement 1).*

For all of our experiments, the light intensity and duration is present on the graphs or in the figure legends. Unless mentioned otherwise, we used a ‘daylight’-like light in order to determine the role of each photoreceptor as shown in (Figure 1—figure supplement 1).

We have recently published a paper in IOVS (Alam et al. IOVS2015) showing that the sensitivity of the different mutants to light are unperturbed in the different mutants when used for object tracking. It also shows that apart from our lowest light intensities (where only rods are activated), the intensities at which cones could not drive pupillary light reflex were sufficient for cones to drive image vision. Therefore, the strength of the stimulus is not the reason that cones are not able to drive the pupillary light reflex.

*The model of pupil constriction (Figure 1—figure supplement 2) has no theoretical or mechanistic basis and, I think, detracts from the paper. Measurement of the consensual pupil reflex with the illuminated eye dilated or not should suffice. I assume this is the experiment shown in Figure 1—figure supplement 2 (the legend does not state whether only one eye was illuminated). If so, the lack of an effect is surprising because the intensity used, 10 lux, should produce substantial constriction of the illuminated eye. Additional discussion is required. Could this experiment be repeated at different light intensities, for direct comparison to the plots in panel C?*

Before we explain this comment, we would like to make it clear to the reviewer and the editor that we are not opposed to removing this figure. However, we believe that once we explain it well, its value will be clearer to this reviewer.

The model presented in Figure 1—figure supplement 2 is based on our experimental data and the fact that light will make the pupil smaller and hence the amount of light going into the retina less. Since the pupil is a circle, then reduction of the circle size will lead to a corresponding decrease in the light going into the retina. In fact, a pupil size of 0.1 would decrease the amount of light by an order of magnitude. Using this information, we mathematically determined an upper bound on negative-feedback’s contribution to PLR decay. The model shows that negative-feedback contributes relatively little to observed PLR decay, at most.

To test this, we did the experiment shown in Figure 1—figure supplement 2 (as mentioned by the reviewer). We applied atropine or PBS only to the left eye and then measured sustained PLR in response to overhead light (10 lux). Because we are stimulating both eyes, we should observe 50% of the impact of negative feedback on the PLR. However, we observed no differences between atropine and PBS, suggesting that negative-feedback plays little to no role in PLR decay in agreement with our model’s predictions. It is important to note that it would have been ideal to measure sustained contralateral PLR for these experiments. But this is not possible since this would require anesthesia to immobilize the mouse for prolonged time and anesthesia is known to alter pupillary responses drastically.

Finally, we chose to use the 10 lux light intensity because it is close to the EC50. Using higher or lower light intensities will not be as efficient in testing this model since we will be hitting either small pupillary light responses or saturated pupillary light responses.